# FireSonic: Design and Implementation of an Ultrasound Sensing-Based Fire Type Identification System

**DOI:** 10.3390/s24134360

**Published:** 2024-07-05

**Authors:** Zheng Wang, Yanwen Wang, Mingyuan Liao, Yi Sun, Shuke Wang, Xiaoqi Sun, Xiaokang Shi, Yisen Kang, Mi Tian, Tong Bao, Ruiqi Lu

**Affiliations:** 1College of Electrical and Information Engineering, Hunan University, Changsha 410012, China; 2China Electric Power Research Institute, Beijing 100192, China; 3State Grid Hunan Electric Power Company Limited, Changsha 410004, China; 4School of Computer Science and Engineering, Central South University, Changsha 410017, China

**Keywords:** acoustic sensing, channel measurements, fire type classification, beamforming

## Abstract

Accurate and prompt determination of fire types is essential for effective firefighting and reducing damage. However, traditional methods such as smoke detection, visual analysis, and wireless signals are not able to identify fire types. This paper introduces FireSonic, an acoustic sensing system that leverages commercial speakers and microphones to actively probe the fire using acoustic signals, effectively identifying fire types. By incorporating beamforming technology, FireSonic first enhances signal clarity and reliability, thus mitigating signal attenuation and distortion. To establish a reliable correlation between fire type and sound propagation, FireSonic quantifies the heat release rate (HRR) of flames by analyzing the relationship between fire-heated areas and sound wave propagation delays. Furthermore, the system extracts spatiotemporal features related to fire from channel measurements. The experimental results demonstrate that FireSonic attains an average fire type classification accuracy of 95.5% and a detection latency of less than 400 ms, satisfying the requirements for real-time monitoring. This system significantly enhances the formulation of targeted firefighting strategies, boosting fire response effectiveness and public safety.

## 1. Introduction

Fire is one of the most significant global threats to life and property. Annually, fire incidents result in billions of dollars of property damage and a substantial number of casualties, such as those from electrical and kitchen fires [1]. Different types of fire exhibit distinct combustion characteristics and spread rates, which require specific firefighting methods [2]. Accurately identifying the type of fire is crucial for public safety. Timely and effective determination of the fire type enables the implementation of targeted firefighting strategies, thereby minimizing fire-related damage and safeguarding lives and property. The study of different types of fires enables a deeper understanding of their combustion mechanisms and development processes. This knowledge facilitates the development of more scientific and effective firefighting techniques and equipment [3]. Not only does this enhance firefighting efficiency, but it also provides essential theoretical foundations and practical guidelines for fire prevention.

With advances in sensor technology, the development of new types of sensing systems has become possible [4,5]. Existing fire detection methods primarily rely on smoke [6,7,8,9], vision [10,11,12,13,14], radio frequency [15,16,17], and acoustics [18,19,20,21,22]. However, each of these methods has its limitations. Smoke detectors, while widely used, are prone to false alarms triggered by environmental impurities and may pose health risks due to the emitted radioactive materials they contain [9,23]. Vision-based detection systems require an unobstructed line of sight and may raise privacy concerns. Furthermore, as illustrated in Figure 1, vision-based methods struggle when cameras are contaminated or obscured by smoke, making it difficult to gather relevant information about the fire’s severity and type. In contrast, methods using wireless signals, such as RF and acoustics, can still capture pertinent data, even in smoky conditions. Recent advances have seen RF sensing technology employed in fire detection tasks. However, RF detection typically requires specialized and expensive equipment and may be inaccessible in remote areas [15,17].

Acoustic-based methods for determining fire types offer several advantages [24,25,26,27,28,29,30,31]. Firstly, they provide a non-contact detection method, which avoids direct contact with the fire. Second, they enable real-time monitoring and rapid response, facilitating timely firefighting measures. Additionally, this method demonstrates strong environmental adaptability, being capable of working well in environments with smoke diffusion or low visibility, providing a more reliable means for fire monitoring [29]. However, the state-of-the-art acoustic-based sensing system in [19] has a limited sensing range, is vulnerable to daily interferences, and is not capable of classifying fire types.

To overcome the drawbacks of the aforementioned approaches, this paper introduces FireSonic, an acoustic-based system capable of discriminating fire types, characterized by a low false alarm rate, effective signal detection in smoke-filled environments, rapid detection speed, and low cost. Unlike conventional reflective acoustic sensing technologies [24,25,26,27,28,32,33,34,35], FireSonic operates by detecting changes in sound waves as they pass through flames, thereby offering a novel approach to acoustic sensing technology. This method enhances the practical application and reliability of acoustic sensors in fire detection scenarios. As shown in Figure 2, it displays the curve of the standard temperature over time for burning with sufficient oxygen, including four stages: initial growth, fire growth, full development, and decay. Due to the temperature variations exhibited by different types of fires at various stages, the objective of this study is to develop a method utilizing acoustic means to monitor the changes in this curve, thereby facilitating the determination of fire type. Research on this technology may even provide technical insights into aspects such as the rate of fire spread in forest fires in the future [36].

However, implementing such a system is a daunting task. The first challenge is obtaining high-quality signals for monitoring fires. The fire environment poses a significant challenge due to its complexity, with factors such as smoke, flames, thermal radiation, and combustion by-products interfering with sound wave propagation, resulting in signal attenuation and distortion. In addition, diverse materials like building materials and furniture affect sound waves differently, causing signal reflection, refraction, and scattering, further complicating accurate signal interpretation.

To address the first challenge, we utilize beamforming technology, adjusting the phase and amplitude of multiple sensors to concentrate on signals from specific directions, thereby reducing interference and noise from other directions. By aligning and combining signals from these sensors, beamforming enhances target signals, improving the signal-to-noise ratio, clarity, and reliability. In addition, we employ spatial filtering techniques to mitigate multipath effects caused by sound waves propagating through different media in a fire, thereby minimizing signal distortion and enhancing detection accuracy.

The second challenge lies in establishing a correlation between fire types and acoustic features. Traditional passive acoustic methods are influenced by the varying acoustic signal characteristics generated by different types of fire. Factors such as the type of burning material, flame size, and shape result in diverse spectral and amplitude features of sound waves, making it difficult to accurately match acoustic signals with specific fire types. Moreover, while the latest active acoustic sensing methods can detect the occurrence of fires, the inability of flames to reflect sound waves poses a challenge in associating active acoustic signals with fire types using conventional reflection approaches.

To address the second challenge, different from directly establishing the association between fire types and sound waves, we propose a heat release-based monitoring scheme; particularly, quantifying the correlation between the region of fire heat release and sound propagation delays. This quantification is based on the fact that sound speed increases with temperature. Larger flames generate broader high-temperature areas. We discern the fire type based on real-time changes in heat production during the combustion process.

In a nutshell, our main contributions are summarized as follows:We address the shortcomings of current fire detection systems by incorporating a critical feature into fire monitoring systems. To the best of our knowledge, FireSonic is the first system that leverages acoustic signals for determining fire types.We employ beamforming technology to enhance signal quality by reducing interference and noise, while our flame HRR monitoring scheme utilizes acoustic signals to quantify the correlation between a fire’s heat release regions and sound propagation delays, facilitating fire type determination and accuracy enhancement.We implement a prototype of FireSonic using low-cost commodity acoustic devices. Our experiments indicate that FireSonic achieves an overall accuracy of 95.5% in determining fire types (experiments were conducted with the presence of professional firefighters and were approved by the Institutional Review Board).

## 2. Background

### 2.1. Heat Release Rate

The relationship between the heat release rate (HRR) and acoustic sensing can be derived through a series of physical principles. The relationship between sound speed (*c*) and temperature (*T*) in air can be expressed by the following formula [37]:(1)$$c=c_{0}\sqrt{\frac{T}{T_{0}}}$$
where $c_{0}$ is the sound speed at a reference temperature $T_{0}$ (usually 293 K), which is about 20 °C, approximately 343 m/s.

The HRR is a measure of the energy released by a fire source per unit time, typically expressed in kilowatts (kW). It can be indirectly calculated by measuring the rate of oxygen consumption by the flames. To establish the relationship between HRR and acoustic sensing, we need to consider how thermal energy affects the propagation of sound waves. As the fire releases heat, the surrounding air heats up, leading to an increase in the sound speed in that area. This change in sound speed can be perceived through changes in the propagation time of sound waves detected by sensors.

### 2.2. Channel Impulse Response

As signals traverse through the air, they travel various paths due to reflections from objects in the environment, posing challenges for accurately capturing fire-related information. Consequently, the receiver picks up multiple versions of the signal, each with its own unique delay and attenuation, a phenomenon known as multipath propagation, which can lead to signal distortion and errors in fire monitoring tasks [27,30,38,39,40]. To address this issue, we utilize channel impulse response (CIR) to segment these multipath signals into discrete segments or taps, enabling more precise identification of signals influenced by fire incidents.

To capture the CIR, a predefined frame is transmitted to probe the channel, subsequently received by the receiver. In the case of a complex baseband signal, measuring the CIR involves computing the cyclic cross-correlation between the transmitted and received baseband signals as
(2)$$h\left[\tau \right]=\frac{1}{N}\sum_{n=1}^{N}T\left[n\right]\cdot R^{*}\left[n-\tau \right]$$
where h[τ] is CIR measurement, T[n] signifies the transmitted complex baseband signal, $R^{*}\left[n\right]$ denotes the conjugate of R[n] at the receiver, and τ corresponds to the multipath delays. *N* is the length of T[n]. h[τ] acts as a matrix illustrating the attenuation of signals with varying delays τ over time *t*.

### 2.3. Beamforming

In complex multipath channels, directing signal energy towards specific points of interest minimizes noise and interference, thereby enhancing the target signal. Beamforming involves steering sensor arrays’ beams to predefined angular ranges to detect signals from various directions [41]. This technique is crucial when signals originate from multiple directions and require sensing. Specifically, beamforming selectively amplifies or suppresses incoming signals by controlling the phase and amplitude of sensor array elements. To generate a beam pattern, signal amplification is focused in a desired direction while attenuating noise and interference from other directions, achieved by applying a set of weights to each sensor’s signal. These weights are optimized based on the signal-to-noise ratio (SNR) or desired radiation pattern.

## 3. System Design

### 3.1. Overview

The system overview is outlined in Figure 3. The CIR estimator sequentially measures the transmitted and received Zadoff–Chu (ZC) sequence. Specifically, the ZC sequence is utilized as a probing signal for its good auto-correlation characteristics [27], while a cross-correlation-based approach is adopted for estimating the CIR affected by fire combustion. Based on the captured CIR, the system assesses the occurrence of fire by considering the differences in sound velocity induced by temperature changes and the acoustic energy absorption effects, and it triggers alarms as needed. Subsequently, beamforming is applied to pinpoint the direction of the fire, producing high-quality thermal maps of the fire. Finally, based on the captured patterns of heat release rate features, a classifier is used for fire type identification.

### 3.2. Transceiver

The high-frequency signal generation module utilizes a root Zadoff–Chu sequence as the transmission signal. The root ZC sequence has a length of 128 samples with an auto-correlation coefficient using u = 64 and Nzc = 127. Different settings of u and Nzc parameters are suited for signal output requirements under various bandwidths. The settings here are designed to meet the limitations of human hearing and hardware sampling rates. After frequency domain interpolation, the sequence extends to 2048 points, achieving a detection range of 7 m, and is up-converted to the ultrasonic frequency band. The signal transmission and reception module stores the high-frequency signal in a Raspberry Pi system equipped with controllable speakers and controls playback via a Python program. The signals are received by a microphone array with a sampling rate of 48 kHz.

### 3.3. Signal Enhancement Based on Beamforming

In fire monitoring, beamforming technology plays a crucial role. In complex fire environments, signals often travel multiple paths due to various obstacles, resulting in signal attenuation and interference [41]. To address this issue, we can utilize beamforming technology to steer the receiver’s beam towards specific directions, thereby maximizing the capture of target signals and minimizing noise and interference. As illustrated in Figure 4, in a microphone array with Mic1, Mic2, and Mic3, each microphone receives signals with different delays. By using the signal from Mic2 as a reference and adjusting the phases of Mic1 and Mic3, signals can be aligned, thus concentrating the combined signal energy. By controlling the phase and amplitude of the signals, beamforming technology can selectively enhance signals in the direction of the fire source, thereby improving the sensitivity and accuracy of the fire monitoring system.

### 3.4. Mining Fire-Related Information in CIR

We have described applying beamforming techniques to enhance signals from the direction of the fire while eliminating interference from other directions so far. Next, we focus on extracting information related to the extent of the fire’s impact. As sound waves travel through the high-temperature region, their propagation speed increases. The complex chemical and physical reactions during the fire cause the temperature distribution and shape of the high-temperature region to be uneven, resulting in a non-uniform spatial temperature field distribution. To characterize this distribution, we conduct the following study.

The number of taps is directly related to the spatial propagation distance of the ultrasonic signal. A smaller number of taps indicates a shorter signal propagation distance, while a larger number of taps indicates a longer propagation distance. Therefore, the sensing range *s* can be determined by the number of taps *L*, specifically expressed as s=L×Δd, where $\Delta d=\frac{v}{2f_{s}}$, which represents the distance corresponding to each tap, i.e., the propagation distance of the signal between two adjacent sampling points. $f_{s}$ is the sampling rate of the signal, and *v* is sound speed. Specifically, we conduct CIR measurements frame by frame, where each frame represents a single CIR trace. For ease of visualization, we normalize all signal energy values across the taps to a range between 0 and 255 and represent them using a CIR heatmap, as depicted in Figure 5. The horizontal axis represents time in frames, with hardware devices capturing 23 frames of data per second at a sampling rate of 48 kHz. The vertical axis denotes the sensing range, indicated by acoustical signal delays.

## 4. Results

### 4.1. CIR Measurements before and after Using Beamforming

Figure 6a,b illustrate the results of CIR measurements for an impacted region of 0.3 m in diameter before and after using beamforming, respectively. It can be observed that the thermal image obtained after beamforming is more accurate. This is because beamforming technology allows for a more comprehensive capture of the characteristics of the affected region, thereby enhancing the accuracy and reliability of the measurements. We empirically set a threshold value Th (e.g., Th = 150) to select taps truly affected by high temperatures, i.e., those taps whose signal energy values exceed the threshold. Subsequently, for the selected taps, we sum their energy values and record the sum as the instantaneous heat release rate (HRR) for that frame. It is noteworthy that during the fire combustion process, the HRR closely aligns with the temperature variation trend caused by the fire [42,43]. Therefore, we utilize the HRR to describe the combustion patterns of different fire types.

### 4.2. Fire Type Identification

To achieve a gradual increase in flame size, we arranged four alcohol swabs in a straight line at intervals of 0.2 m within a container, interconnected by a thread soaked in alcohol. These alcohol swabs were sequentially ignited via the thread, illustrating a progressively intensifying fire scenario. The captured CIR measurement results are presented in Figure 7a. From the ignition of the first alcohol swab at the 1st second to the 4th second, the delay difference ΔDelay increased from an initial 1 ms to 4.5 ms. Similarly, by adjusting the valve of a cassette burner, we simulated a scenario of decreasing fire intensity, where the ΔDelay decreased from 3.5 ms to 1.5 ms (Figure 7b). Our experimental results indicate that the delay difference ΔDelay is an effective indicator of fire severity and can be utilized to predict the fire’s propagation tendency.

Each trace represents a frame, and the affected area within each frame changes continuously, thereby facilitating the determination of the fire type. To accurately assess the affected areas caused by different types of fires, we define the region between the two farthest taps in each frame with normalized values greater than the set threshold as the area affected by the fire. By summing the taps contained in the high-temperature region (HTR) of each frame, we obtain the energy variation for describing the heat release rate curve. Subsequently, moving average (MA) processing is applied to reduce short-term fluctuations in the scatter plot, enabling the identification and analysis of the long-term development trends of fires. Next, we conduct detailed studies on different fire types. The moving average technique is employed to smooth the data, which helps eliminate short-term fluctuations and highlight long-term trends. This method is particularly useful because the sound energy influenced by the flames fluctuates significantly between adjacent frames. By calculating the average value over a period, we capture the variations in fire intensity over that period, allowing us to match the characteristics of different fire types accordingly [44].

A fuel-controlled fire occurs when the development and spread of the fire are primarily determined by the quantity and arrangement of the combustible materials, given an ample supply of oxygen. In our experiment, we placed a suitable amount of paper and charcoal into a metal box as fuel, with the container top left open to provide sufficient oxygen, simulating the conditions of a real fuel-controlled fire. The experimental results are shown in Figure 8a. Analysis of the experimental curve reveals that with ample oxygen, the HRR increases rapidly between frames 50 and 100, then stabilizes at around 140 kW. However, at frame 220, the HRR rapidly decreases due to the reduction in combustibles.

A ventilation-limited fire is characterized by the fire’s development being primarily dependent on the amount of oxygen supplied, rather than the amount of fuel. This type of fire typically occurs in enclosed or semi-enclosed spaces. In our simulation, we placed sufficient fuel in the container as fuel and covered the top with a thin metal lid to simulate the oxygen supply conditions of a real ventilation-limited fire. The experimental results are shown in Figure 8b. Analysis indicates that even with ample fuel, the limitation of oxygen causes the HRR to gradually increase, peaking at frame 180 with an HRR of approximately 70 kW, and then, gradually decreasing.

A ventilation-induced flashover fire, also known as backdraft, occurs when flames in an oxygen-deficient environment produce a large amount of incompletely burned products. When suddenly exposed to fresh air, these combustion products ignite rapidly, causing the fire to flare up violently. In our experiment, we initially placed sufficient charcoal and paper in the container as fuel, and covered the top with a thin metal lid to simulate the conditions of oxygen deficiency. As the flame within the container diminished, we used a metal wire to open the lid and introduce a large amount of oxygen, simulating the real-life oxygen supply during a flashover. The HRR curve shown in Figure 8c indicates that in the initial oxygen-deficient phase (before frame 230), the HRR increases, and then, decreases, remaining below 80 kW. After introducing sufficient oxygen by opening the lid, the HRR rapidly increases, reaching a peak of around 90 kW at frame 450, before gradually decreasing.

A pulsating fire refers to a fire where the intensity of combustion fluctuates rapidly during its development, creating a pulsating phenomenon in the combustion intensity. This is caused by instability in the oxygen supply. In our experiment, we placed an appropriate amount of paper and charcoal in the container as fuel, and used a metal lid to control the oxygen supply. After the fire intensified, we repeatedly moved the metal lid back and forth with a metal wire to simulate the instability of oxygen supply during a real fire. As shown in Figure 8d, the HRR gradually increases to around 85 kW when both fuel and oxygen are sufficient. After frame 250, the instability in oxygen supply, caused by moving the metal lid, leads to the HRR fluctuating around 80 kW.

### 4.3. Classifier

We chose support vector machine (SVM) as the classifier for fire type identification, with the smoothed curves as inputs. The fundamental principle is to find an optimal hyperplane that maximizes the margin between sample points of different fire classes. This approach of maximizing margins enhances the generalization capability of the model, leading to more accurate classification of new samples. Moreover, SVM focuses mainly on support vectors rather than all samples during model training. This sensitivity to support vectors enables SVM to effectively address classification issues in imbalanced datasets.

## 5. Experiments and Evaluation

### 5.1. Experiment Setup

#### 5.1.1. Hardware

The hardware of our system is depicted in Figure 9a, consisting of a commercial-grade speaker and a circular arrangement of microphones. The acoustic signals emitted by the speaker are partially absorbed by the flames before being captured by the microphones. The hardware components utilized in our system are depicted in Figure 9a, comprising a commercial-grade speaker and a circular arrangement of microphones. To facilitate the precise transmission of acoustic signals, we employ the Google AIY Voice Kit 2.0, which includes a 3 W speaker controlled by an efficient Raspberry Pi Zero. For the reception of signals, we selected the ReSpeaker 6-Mic Circular Array Kit that operates at a sampling rate of 48 kHz, ensuring the accurate capture of audio data in the inaudible frequency range. The FireSonic system costs less than USD 30. The Raspberry Pi is developed and promoted by the Raspberry Pi Foundation, headquartered in Cambridge, UK.

#### 5.1.2. Data Collection

We configured the root ZC sequence with the parameters u=64 and $N_{zc}=127$, and then, upscaled it to a sequence length of ${N\prime }_{ZC}=2048$ to ensure coverage of a 7.25 m sensing range. The data were gathered across three distinct rooms, each with different dimensions: 4 m × 6 m × 3 m, 5 m × 5 m × 3 m, and 7 m × 8 m × 3 m. These room layouts are depicted in Figure 9b–d. A total of 1000 fire type measurements were conducted in each room, resulting in a total of 3000 complete measurements.

#### 5.1.3. Model Training, Testing, and Validation

We utilize the HRR curve after MA as the model’s input, providing ample information for fire monitoring. For the offline training process, we leverage a computing system equipped with 32 GB of RAM (NVIDIA Corporation, Santa Clara, CA, USA), an Intel Core i7-13700K CPU (Intel Corporation, Santa Clara, CA, USA) from the 13th Generation lineup, and an NVIDIA GeForce RTX 4070 GPU (NVIDIA Corporation, Santa Clara, CA, USA). Following training, the model is stored on a Raspberry Pi for subsequent online testing. The trained model has a file size of less than 6 megabytes. It is worth noting that we also conducted tests on unfamiliar datasets to further validate the system’s performance.

### 5.2. Evaluation

#### 5.2.1. Overall Performance

We name the four types of fires corresponding to Figure 8 as type 1, type 2, type 3, and type 4, respectively. Figure 10a shows the system’s judgment results on the four types of fires in three rooms, achieving an average recognition accuracy rate of 95.5%. As shown in Figure 10b, even in a large room like room 3, when a fire breaks out the accuracy rate can reach above 94.5%. With our holistic design, the system is able to provide accurate fire state information to firefighters, allowing them to take appropriate countermeasures.

#### 5.2.2. Performance on Different Classifiers

To better demonstrate the effectiveness of capturing fire type features, we compared SVM with BP neural network (BP), random forest (RF), and k-nearest neighbors (KNN), as illustrated in Table 1. SVM exhibited the best performance in terms of highest identification accuracy, reaching 97.6%. The overall data demonstrate high discriminability. However, the introduction of nonlinearity by BP neural network did not effectively leverage its advantages. RF and KNN were significantly influenced by outliers and noise, leading to inferior performance compared to SVM.

#### 5.2.3. Performance in Different Locations

We randomly selected nine locations in room 3, as shown in Figure 10c. We obtained the average classification accuracy for each location, and all were above 93.3%. This performance validates the rationality of our technique in capturing fire type differences from various angles and distances, as well as corroborates the judiciousness of our feature selection based on smoothing via moving averages. This substantiates the efficacy and robustness of our holistic design for accurate fire situation awareness under diverse scenarios.

#### 5.2.4. Performance in Smoky Environments

We randomly selected three locations within room 1 to simulate the performance in a smoke-filled fire scenario. The system’s performance, as shown in Figure 10d, indicates that FireSonic can achieve an accuracy rate of over 94% even in complex smoke environments. This high level of accuracy is due to the system fundamentally operating on the principle of monitoring heat release.

#### 5.2.5. Performance on Different Fuels

To address potential biases associated with testing using a single fuel source, we expanded our evaluation to include six diverse fuels: ethanol, paper, charcoal, wood, plastics, and liquid fuel. The results, as presented in Table 2, demonstrate that the system consistently maintains an average accuracy exceeding 94% across three different locations and all fuel scenarios. This uniform high accuracy confirms that the fire evolution features captured by the system are both discriminative and robust. Furthermore, these outcomes affirm the system’s ability to generalize effectively across a variety of fuel types, thereby underscoring the design’s overall effectiveness in providing precise fire situation awareness.

#### 5.2.6. Comparison of Performance with and without Beamforming

To further investigate the extent of performance enhancement by beamforming, we conducted 300 measurements each with and without beamforming using FireSonic in room 3. The results, as depicted in Figure 11, show that precision, recall, and F1-score values are all above 95% when beamforming is employed. In stark contrast, experiments without beamforming show performance below 85%. This confirms the beneficial role of beamforming in capturing fire-related features via CIR.

#### 5.2.7. Detection Time

The system detection time includes a frame detection time of less than 200 ms, a downconversion time of 20 ms, a CIR measurement time of 30 ms, and a fire type judgment computation time of 130 ms. The cumulative detection duration of these four steps is less than 400 ms, which sufficiently meets the real-time requirements in urgent fire scenarios.

#### 5.2.8. Comparison with State-of-the-Art Work

To benchmark against state-of-the-art acoustic-based fire detection approaches, we employ the system in [19] as a baseline, which also leverages acoustic signals for fire detection. As illustrated in Table 3, the comparison highlights significant differences in detection capabilities. Current advanced technologies can only effectively detect fire-related sounds within a range of less than 1 m. In contrast, FireSonic extends this detection range to seven meters, significantly broadening the scope of detection. We also evaluated the accuracy of the system under conditions where the range is less than one meter, and FireSonic achieves an average accuracy of 98.7%, which is 1.4% higher than the most recent technologies. Existing technologies show varying sensitivities to the acoustic characteristics of different materials, often performing poorly with certain types. Conversely, FireSonic primarily detects flames through monitoring heat release, thus adapting to a wider variety of materials and ignition scenarios, more closely aligning with the real demands of fire incident contexts. Additionally, FireSonic uses ultrasonic waves to detect flames, ensuring it is not affected by common low-frequency noise in the environment, such as conversations, thereby exhibiting enhanced noise immunity. Moreover, FireSonic includes the capability to identify types of fires, a significant advantage that current technologies do not offer.

## 6. Conclusions

In this study, we developed and validated FireSonic, an innovative ultrasound-based system for accurately identifying indoor fire types. Key achievements include an extended sensing range of up to 7 m, high classification accuracy of 95.5%, robust performance across various materials, resistance to environmental noise, real-time monitoring capabilities with a detection latency of less than 400 ms, and the ability to identify the fire type. By incorporating beamforming technology, FireSonic enhances signal quality and establishes a reliable correlation between heat release and sound propagation. These advancements demonstrate FireSonic’s significant scientific and practical contributions to fire detection and safety.

## Figures and Tables

**Figure 1 sensors-24-04360-f001:**
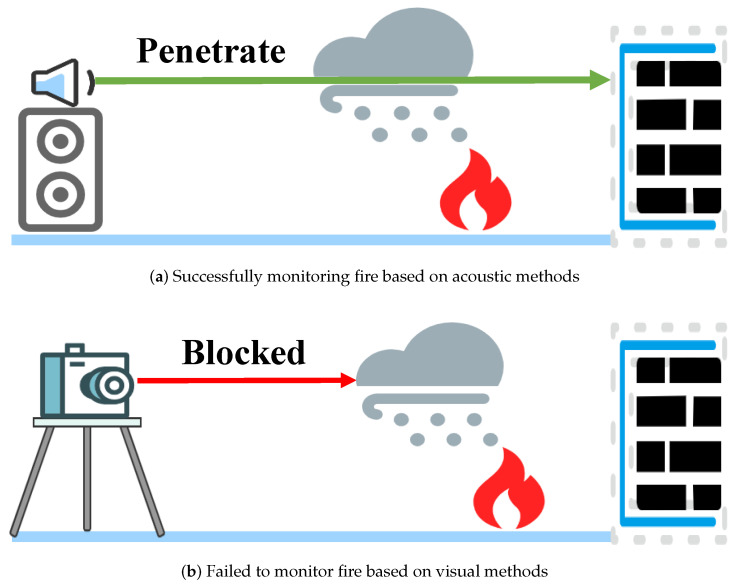
Monitoring in the presence of smoke.

**Figure 2 sensors-24-04360-f002:**
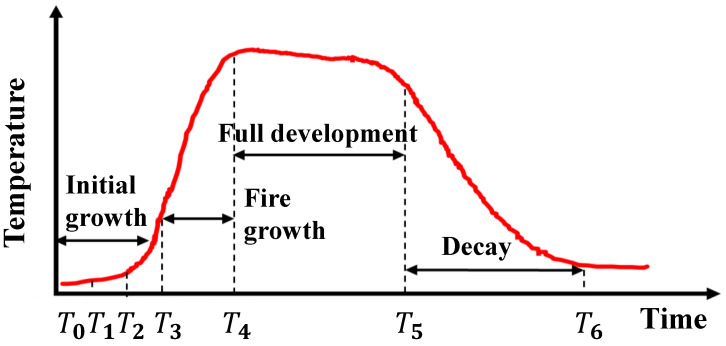
Fire evolution.

**Figure 3 sensors-24-04360-f003:**
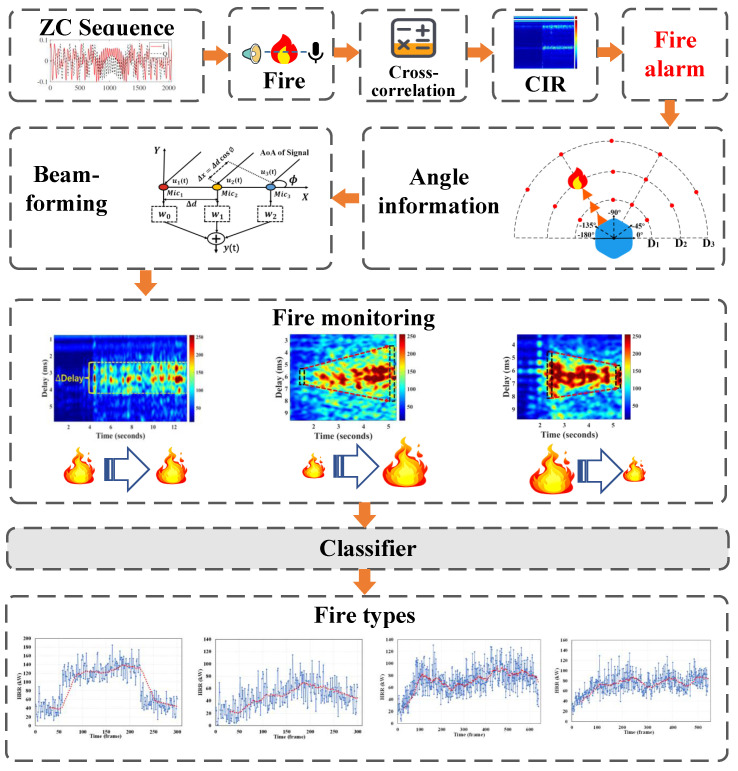
Overview.

**Figure 4 sensors-24-04360-f004:**
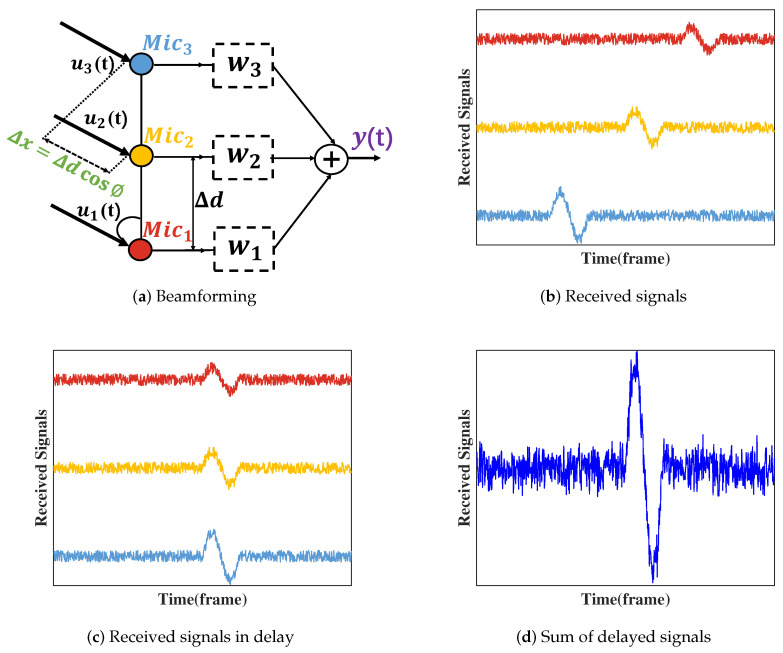
Beamforming.

**Figure 5 sensors-24-04360-f005:**
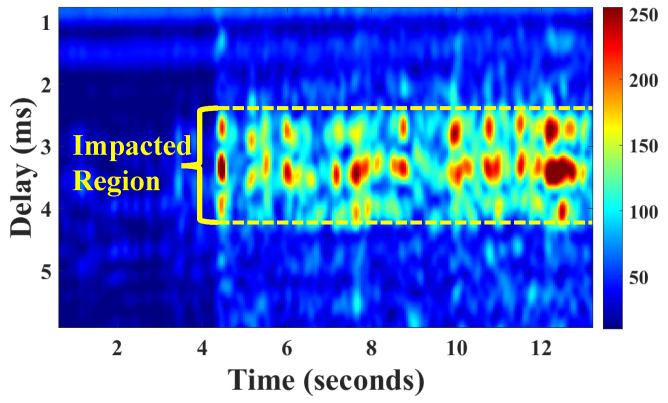
High-temperature region in CIR measurement.

**Figure 6 sensors-24-04360-f006:**
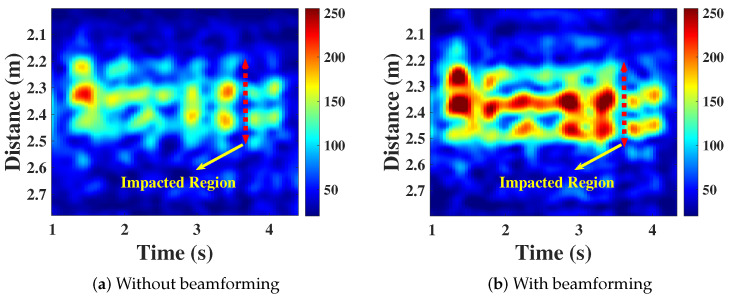
Comparison with and without using beamforming.

**Figure 7 sensors-24-04360-f007:**
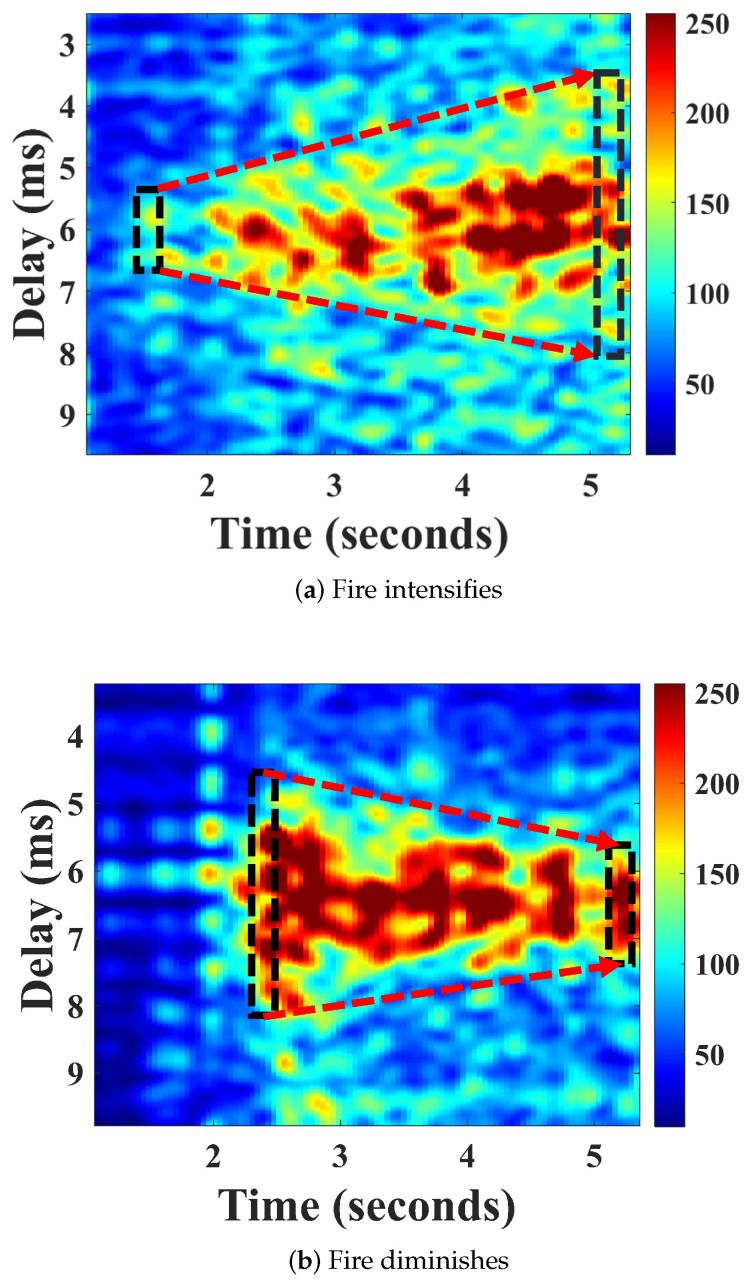
Fire monitoring in CIR.

**Figure 8 sensors-24-04360-f008:**
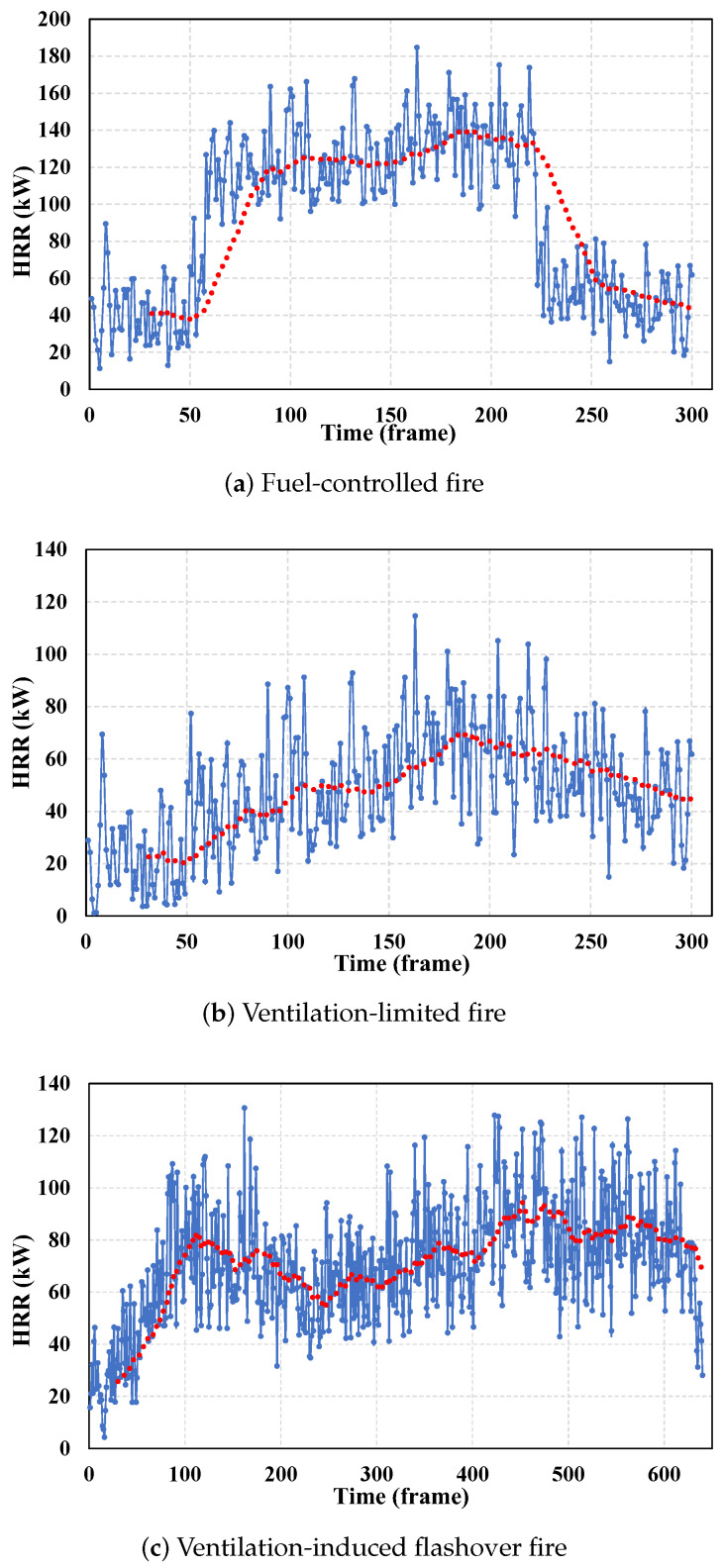
Combustion curves of different fire types.

**Figure 9 sensors-24-04360-f009:**
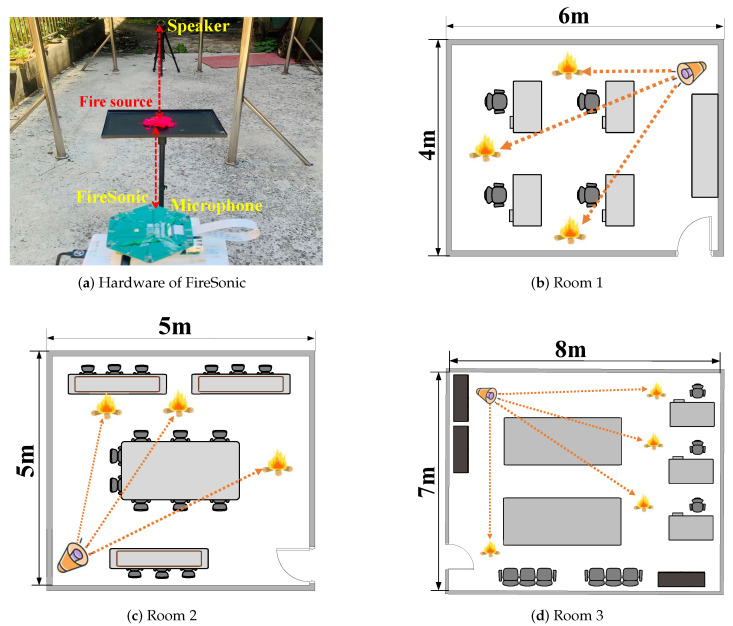
Commercial speaker and microphone applied in our system and three rooms.

**Figure 10 sensors-24-04360-f010:**
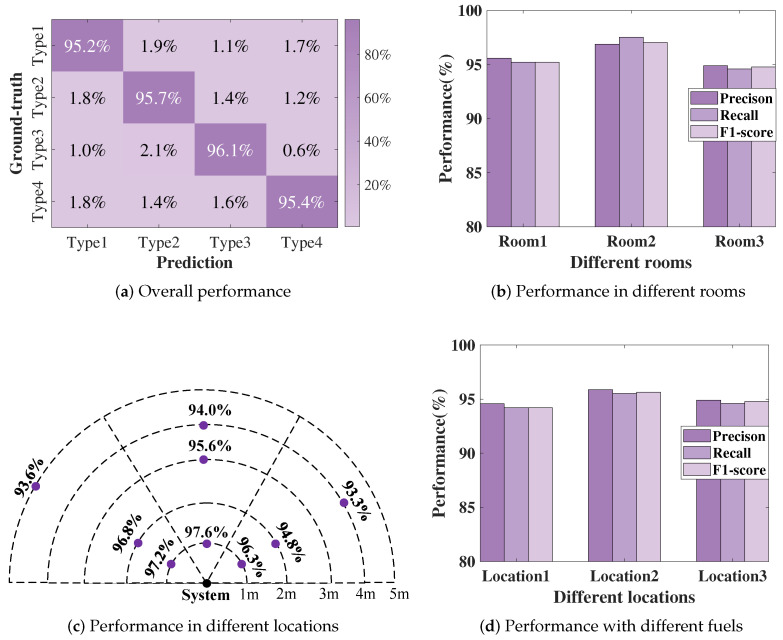
System performance.

**Figure 11 sensors-24-04360-f011:**
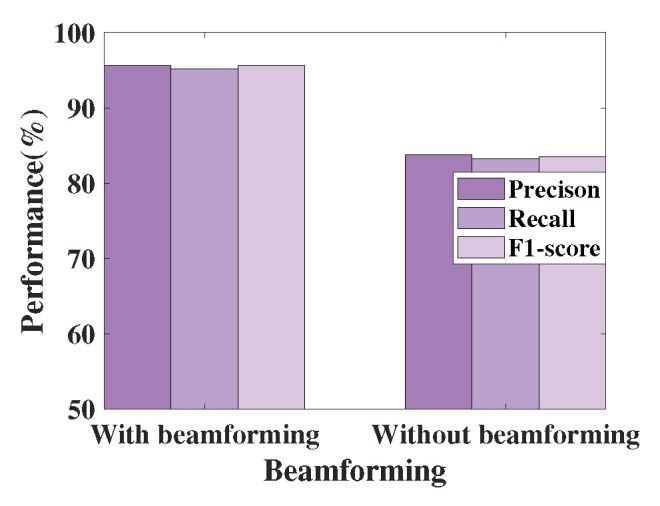
Comparison of performance with and without beamforming.

**Table 1 sensors-24-04360-t001:** Performance on different classifiers.

Classifier	SVM	BP	RF	KNN
**Best performance**	97.6%	95.2%	89.3%	75.9%
**Worst performance**	93.3%	90.3%	78.5%	71.5%
**Average performance**	95.5%	93.5%	84.7%	72.8%

**Table 2 sensors-24-04360-t002:** Performance with different fuels.

Type of Fuel	Ethanol	Paper	Charcoal	Woods	Plastics	Liquid Fuel
**Performance**	97.2%	98.6%	95.1%	94.3%	98.2%	97.5%

**Table 3 sensors-24-04360-t003:** Comparison with State-of-the-art Work.

	Sensing Range	Accuracy (1 m)	Performance Across Different Materials	Resistance to Daily Interference	Ability to Classify Fire Types
**FireSonic**	7m	98.7%	Varies minimally	Strong	Yes
**State-of-** **the-art Work**	1m	97.3%	Varies significantly	Weak	No

## Data Availability

The data used to support the findings of this study are available from the corresponding author upon request.

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
