# Peer review of "FireSonic: Design and Implementation of an Ultrasound Sensing-Based Fire Type Identification System"

_sensors, 2024, doi:10.3390/s24134360_

Round 1
Reviewer 1 Report
Comments and Suggestions for Authors
This paper introduces the design and implementation of a fire type identification system based on ultrasonic sensing called FireSonic. FireSonic uses commercial speakers and microphones to actively detect the source of fire through acoustic signals and effectively identify indoor fire types. The system uses beamforming technology to enhance signal clarity and reliability, and reduce signal attenuation and distortion. In addition, FireSonic quantifies the heat release rate (HRR) of the flame by analyzing the relationship between the heated region of the fire source and the propagation delay of the sound wave, and extracts the spatiotemporal characteristics associated with the fire from the channel measurements.The article has a certain innovation. This article still needs to be improved, my comments are as follows:
1. Advantages over traditional methods are mentioned in the paper, but direct comparisons with existing acoustic wave sensing techniques are lacking. It is suggested that the authors provide more comparative experiments to prove the superiority of the FireSonic system in practical applications. The validation section should include tests of different environmental conditions and fire scenarios to demonstrate the robustness of the system.
2. By combining beamforming techniques and heat release rate monitoring, the method demonstrated high classification accuracy and low latency in experiments. However, it is recommended that the authors further elaborate on the technical advantages over existing methods, such as how FireSonic performs in complex environments (such as smoky haze) compared to traditional smoke detection or visual detection methods, as well as its specific advantages in terms of real-time and cost effectiveness.
3. This paper describes briefly the development process of the FireSonic system, including the acquisition, processing and analysis of acoustic signals. The experimental design is reasonable, the procedure is clear and the method is innovative. However, the description of the acoustic signal processing algorithm is relatively brief, and it is suggested that the authors add more technical details, such as the signal processing flow, algorithm parameter selection and optimization process. This will help readers to better understand and reproduce the research work.
Comments on the Quality of English LanguageMinor recvision.
Author Response
Comments 1: Advantages over traditional methods are mentioned in the paper, but direct comparisons with existing acoustic wave sensing techniques are lacking. It is suggested that the authors provide more comparative experiments to prove the superiority of the FireSonic system in practical applications. The validation section should include tests of different environmental conditions and fire scenarios to demonstrate the robustness of the system.
Response 1: Thank you for your insightful feedback on our manuscript. We agree with this comment. Therefore, we made comprehensive comparisons across five key dimensions: sensing range, accuracy, performance across different materials, resistance to daily interference, and the ability to classify fire types as shown in the new added Table 3. These changes aim to more effectively highlight the advantages of FireSonic over current technologies. These revisions have been marked in red to easily distinguish the updates. The revised content can be found in Page 14, Paragraph 3. The modified content is as follows:
5.2.8 Comparison with State-of-the-art Work
To benchmark against state-of-the-art acoustic-based fire detection approaches, we employ the system in [19] as a baseline, which also leverages acoustic signals for fire detection. As illustrated in Table3, the comparison highlights significant differences in detection capabilities. Current advanced technologies can only effectively detect fire-related sounds within a range of less than 1m. In contrast, FireSonic extends this detection range to seven meters, significantly broadening the scope of detection. We also evaluated the accuracy of the system under conditions where the range is less than one meter, and FireSonic achieves an average accuracy of 98.7%, which is 1.4% higher than the most recent technologies. Existing technologies show varying sensitivities to the acoustic characteristics of different materials, often performing poorly with certain types. Conversely, FireSonic primarily detects flames through monitoring heat release, thus adapting to a wider variety of materials and ignition scenarios, more closely aligning with the real demands of fire incident contexts. Additionally, FireSonic uses ultrasonic waves to detect flames, ensuring it is not affected by common low-frequency noise in the environment, such as conversations, thereby exhibiting enhanced noise immunity. Moreover, FireSonic includes the capability to identify types of fires, a significant advantage that current technologies do not offer.
We hope such an explanation and revision could address the reviewer’s concern.
Comments 2: By combining beamforming techniques and heat release rate monitoring, the method demonstrated high classification accuracy and low latency in experiments. However, it is recommended that the authors further elaborate on the technical advantages over existing methods, such as how FireSonic performs in complex environments (such as smoky haze) compared to traditional smoke detection or visual detection methods, as well as its specific advantages in terms of real-time and cost effectiveness.
Response 2: Thank you for pointing this out. We agree with this comment. Therefore, we expand our introdution in the manuscript to highlight how FireSonic excels in complex environments, such as smoke, compared to traditional smoke detectors and visual detection systems. Specifically, we detail FireSonic's robust performance in environments where visual and traditional sensors may fail due to obscurity or false alarms triggered by non-fire-related smoke. We also elaborate on FireSonic's real-time processing capabilities and cost-effectiveness, demonstrating its advantages over conventional systems which often require more complex and expensive infrastructure. The revised content can be found in Page 2, Paragraph 3 and Page 13, Paragraph 3. The modified contents are as follows:
[Page 2, Paragraph 3:
To overcome the drawbacks in aforementioned approaches, this paper introduces FireSonic, an acoustically-based system capable of discriminating fire types, characterized by a low false alarm rate, effective signal detection in smoke-filled environments, rapid detection speed, and low cost. Unlike conventional reflective acoustic sensing technologies [24-28,37-40], FireSonic operates by detecting changes in sound waves as they pass through flames, thereby offering a novel approach to acoustic sensing technology.]
[Page 13, Paragraph 3:
5.2.4. Performance in smoky environments
We randomly selected three locations within Room 1 to simulate the performance in a smoke-filled fire scenario. The system's performance, as shown in Fig.10(d), indicates that FireSonic can achieve an accuracy rate of over 94% even in complex smoke environments. This high level of accuracy is due to the system fundamentally operating on the principle of monitoring heat release.]
We hope such an explanation and revision could address the reviewer’s concern.
Comments 3: This paper describes briefly the development process of the FireSonic system, including the acquisition, processing and analysis of acoustic signals. The experimental design is reasonable, the procedure is clear and the method is innovative. However, the description of the acoustic signal processing algorithm is relatively brief, and it is suggested that the authors add more technical details, such as the signal processing flow, algorithm parameter selection and optimization process. This will help readers to better understand and reproduce the research work.
Response 3: Thank you for pointing this out. We agree with this comment. Therefore, we thoroughly revised the manuscript to incorporate a more comprehensive explanation of the signal processing workflow and other pertinent technical details. Specifically, we have added a new section titled "Transceiver" to provide a detailed description of the signal processing algorithms. The revised sections have been highlighted in red within the manuscript to ensure they are easily identifiable. The revised content can be found in Page 5, Paragraph 3. The modified content is as follows:
3.2. Transceiver
The high-frequency signal generation module utilizes a root Zadoff-Chu sequence as the transmission signal. The root ZC sequence has a length of 128 samples with an auto-correlation coefficient using u=64 and Nzc=127. Different settings of u and Nzc parameters are suited for signal output requirements under various bandwidths. The settings here are designed to meet the limitations of human hearing and hardware sampling rates. After frequency domain interpolation, the sequence extends to 2048 points, achieving a detection range of 7 meters, and is up-converted to the ultrasonic frequency band. The signal transmission and reception module stores the high-frequency signal in a Raspberry Pi system equipped with controllable speakers and controls playback via a Python program. The signals are received by a microphone array with a sampling rate of 48 kHz.
We hope that these enhancements address your concerns, and we are grateful for your guidance in improving the quality and depth of our manuscript.
The content of the attachment is the same as that filled out on the webpage.

Reviewer 2 Report
Comments and Suggestions for Authors
This paper attempts to identify types of enclosure (building and residential) fires, which are different from open wildland fires. They assumed a reliable correlation to exist between fire types and sound wave propagation. They attempted to quantify the heat release rate of flames by analyzing the relationship between fire heated areas and sound wave propagation delays. They collected experimental data to develop/validate FireSonic model. This paper is worthy of publication after the authors make a successful revision based on the following comments.
(1) Figure 2 shows a schematic of entire fire process, ignition to fire growth to steady burning, and fire decay leading to extinction. Their term “detonation” may not be applicable for most building and residential fires, since it requires pressure-driven supersonic combustion. Please review the definition of detonation and change it to fire growth, for example, if necessary.
(2) In introduction, between lines 71 – 78, they wrote, “To address the second challenge, different from directly establishing the association between fire types and sound waves, we propose a heat release based monitoring scheme, particularly quantifying the correlation between the region of fire heat release and sound propagation delays. This quantification is based on the fact that sound speed increases with temperature elevation. Our observations indicate that larger flames generate broader high-temperature areas, resulting in higher heat release rates. Variations in heat release rates differ among different fire types, thereby providing possibilities for fire type determination.” The last line holds an important assumption, which was not successfully justified in this study, because there is a commonly accepted standardized heat release rate, e.g., 25 kW/m2 in building fires (see e.g., K. Saito, J.G. Quintiere, and F.A. Williams, Fire Safety Science – Proc. First International Symposium, 1986, pp.75 - 86.)
(3) Figure 3 schematic is helpful for readers to effectively capture their concept of acoustic detection of fire.
(4) In the section, Performance on different fuels, they wrote, “To mitigate the potential bias from testing with a single fuel source, we evaluated the system’s performance using three different fuels: ethanol, paper, and charcoal. Satisfactorily, as depicted in Fig. 10(d), the system achieved an average classification accuracy exceeding 94% at three locations across all three fuel scenarios. …. The results validate the system’s capability to generalize across diverse fuel types, further corroborating the efficacy of the holistic design for accurate fire situation awareness.” However, their three different types of fuels do not represent woods, plastics, and liquid fuels, commonly used for fire research, which offers rich literature for them to evaluate their sensor model capability. Their conclusion, “these findings underscore the FireSonic system’s substantial scientific validity and rationality in fire type identification” should be reevaluated after reviewing the above references.
(5) This FireSonic may be applied to measure the fire spreading rate in forest fires, an important parameter in forest firefighting strategy. If it works, this paper becomes more valuable. See Ref. A. Darwish, et al., Fuel 304 (2021) 121371.
Comments on the Quality of English LanguageMinor English editing will make this paper good quality.
Author Response
Comments 1: Figure 2 shows a schematic of entire fire process, ignition to fire growth to steady burning, and fire decay leading to extinction. Their term “detonation” may not be applicable for most building and residential fires, since it requires pressure-driven supersonic combustion. Please review the definition of detonation and change it to fire growth, for example, if necessary.
Response 1: Thank you for pointing this out. We agree with this comment. Regarding the term "detonation," we acknowledge your point that it may not be suitable for describing typical building and residential fires, as detonation implies pressure-driven supersonic combustion, which is not characteristic of these scenarios. We have carefully reviewed the definition of "detonation" and replaced it with a more appropriate term such as "fire growth" as suggested, in the revised manuscript. The revised content can be found in Page 2, Paragraph 3.
We hope such an explanation and revision could address the reviewer’s concern.
Comments 2: In introduction, between lines 71 – 78, they wrote, “To address the second challenge, different from directly establishing the association between fire types and sound waves, we propose a heat release based monitoring scheme, particularly quantifying the correlation between the region of fire heat release and sound propagation delays. This quantification is based on the fact that sound speed increases with temperature elevation. Our observations indicate that larger flames generate broader high-temperature areas, resulting in higher heat release rates. Variations in heat release rates differ among different fire types, thereby providing possibilities for fire type determination.” The last line holds an important assumption, which was not successfully justified in this study, because there is a commonly accepted standardized heat release rate, e.g., 25 kW/m2 in building fires (see e.g., K. Saito, J.G. Quintiere, and F.A. Williams, Fire Safety Science – Proc. First International Symposium, 1986, pp.75 - 86.)
Response 2: Thank you for pointing this out. We agree with this comment. As the reviewer mentioned, "Our observations indicate that larger flames generate broader high-temperature areas, resulting in higher heat release rates. Variations in heat release rates differ among different fire types, thereby providing possibilities for fire type determination." To clarify this statement, we have revised it to: "Larger flames generate broader high-temperature areas. We discern fire types based on real-time changes in heat production during the combustion process.”
The revised content can be found in Page 3, Paragraph 4.
We hope such an explanation and revision could address the reviewer’s concern.
Comments 3: Figure 3 schematic is helpful for readers to effectively capture their concept of acoustic detection of fire.
Response 3: Thank you for your positive feedback on Figure 3. We are pleased to hear that the schematic has been helpful in conveying our concept of acoustic detection of fire to readers.
Comments 4: In the section, Performance on different fuels, they wrote, “To mitigate the potential bias from testing with a single fuel source, we evaluated the system’s performance using three different fuels: ethanol, paper, and charcoal. Satisfactorily, as depicted in Fig. 10(d), the system achieved an average classification accuracy exceeding 94% at three locations across all three fuel scenarios. …. The results validate the system’s capability to generalize across diverse fuel types, further corroborating the efficacy of the holistic design for accurate fire situation awareness.” However, their three different types of fuels do not represent woods, plastics, and liquid fuels, commonly used for fire research, which offers rich literature for them to evaluate their sensor model capability. Their conclusion, “these findings underscore the FireSonic system’s substantial scientific validity and rationality in fire type identification” should be reevaluated after reviewing the above references.
Response 4: We agree with this comment. Thank you for your constructive feedback on our manuscript, particularly regarding the section on "Performance on Different Fuels." We appreciate your insightful comments regarding the representation of fuel types used in our study. In response to your suggestions, we have conducted additional experiments evaluating the system's performance across six different fuel types, including commonly used materials such as wood, plastics, and liquid fuels. The results after evaluation are shown in Table 1. Our updated results demonstrate satisfactory average classification accuracies exceeding 94% across all six fuel scenarios. The revised content can be found in Page 13, Paragraph 4. The modified content is as follows:
5.2.5. Performance on different fules
To address potential biases associated with testing using a single fuel source, we expanded our evaluation to include six diverse fuels: ethanol, paper, charcoal, wood, plastics, and liquid fuel. The results, as presented in Table 1, demonstrate that the system consistently maintains an average classification accuracy exceeding 94% across three different locations and all fuel scenarios. This uniform high accuracy confirms that the fire evolution features captured by the system are both discriminative and robust. Furthermore, these outcomes affirm the system's ability to generalize effectively across a variety of fuel types, thereby underscoring the design's overall effectiveness in providing precise fire situation awareness.
We hope such an explanation and revision could address the reviewer’s concern.
Comments 5: This FireSonic may be applied to measure the fire spreading rate in forest fires, an important parameter in forest firefighting strategy. If it works, this paper becomes more valuable. See Ref. A. Darwish, et al., Fuel 304 (2021) 121371.
Response 5: Thank you for providing valuable feedback regarding the potential application of FireSonic in measuring the fire spreading rate in forest fires, a crucial parameter in forest firefighting strategy. Your insights are greatly appreciated. We have incorporated the references you suggested into our manuscript. Drawing from this literature and our insights gained during the development of FireSonic, we recognize that employing higher power and antenna gain equipment could indeed facilitate the adaptation of FireSonic for forest fire scenarios. Moving forward, we plan to focus on further research into forest fire dynamics as a future direction, aiming to address the broader spectrum of fire detection scenarios more comprehensively. In the introduction section where FireSonic is described, we added content regarding this topic and marked these additions in red. The revised content can be found in the last sentence of Paragraph 3, Page 2. The modified content is as follows:
Research on this technology may even provide technical insights into aspects such as the rate of fire spread in forest fires in the future [41].
Once again, we sincerely appreciate your constructive feedback, which enhances the depth and applicability of our study.

Reviewer 3 Report
Comments and Suggestions for Authors
Dear Authors,
This manuscript presented a FireSonic, an acoustic sensing system leveraging commercial speakers and microphones to actively probe the fire by acoustic signals, recognizing indoor fire types. The manuscript is an appreciated and absorbing study and can be published after considering some minor revisions.
-Self citation should be removed from the introduction. It is better to use some other updated references particularly from the journal. Also, about the sensor, the sensors used in these references can be useful for your study.
https://doi.org/10.1007/s11071-020-05668-6
https://doi.org/10.1016/j.jtbi.2022.111311
-Section 2 should be expanded. A reader needs to know more about the background so that he will be absorbed to continue reading.
-Coherency between sections should be considered.
-Section 4 (experiment setup) should be more obvious for readers.
-A separate section related to results must be added to the manuscript. Please determine it separately.
-Conclusion needs to be rewritten. It should contain main achievements. The authors can use bullet points to clarify them.
-Some English language errors should be addressed.
Regard
Comments on the Quality of English LanguageThere are some English language errors that should be addressed.
Author Response
Comments 1: Self citation should be removed from the introduction. It is better to use some other updated references particularly from the journal. Also, about the sensor, the sensors used in these references can be useful for your study.
https://doi.org/10.1007/s11071-020-05668-6
https://doi.org/10.1016/j.jtbi.2022.111311
Response 1: Thank you for pointing this out. We agree with this comment. We addressed your suggestion by removing self-citations from the introduction and revising the descriptions of state-of-the-art work, as highlighted in red in the third paragraph of Introduction. Additionally, we have carefully reviewed and incorporated the sensors used in the references you mentioned, significantly enhancing the relevance and robustness of our study. These modifications are indicated by the red text in the second paragraph of Introduction. The two modifications are as follows:
[the third paragraph of the Introduction:
However, the state-of-the-art work acoustic-based sensing system in [19] has a limited sensing range, vulnerable to daily interferences and is not capable of classifying fire types.]
[the second paragraph of Introduction:
With advances in sensor technology, the development of new types of sensing systems has become possible[4,5].]
We hope such an explanation and revision could address the reviewer’s concern.
Comments 2: Section 2 should be expanded. A reader needs to know more about the background so that he will be absorbed to continue reading.
Response 2: Thank you for pointing this out. We agree with this comment. We added a new section called “Beamforming” based on your valuable suggestion, as it is a crucial technique for enhancing the accuracy of our system in this paper. This addition not only enriches the research content but also aids readers in gaining a more comprehensive understanding of our study topic. The added content is shown in Section 2.3, Page 4, the last paragraph. The added content is as follows:
2.3. Beamforming
In complex multipath channels, directing signal energy towards specific points of interest minimizes noise and interference, thereby enhancing the target signal. Beamforming involves steering sensor arrays' beams to predefined angular ranges to detect signals from various directions [50]. This technique is crucial when signals originate from multiple directions and require sensing. Specifically, beamforming selectively amplifies or suppresses incoming signals by controlling the phase and amplitude of sensor array elements. To generate a beam pattern, signal amplification is focused in a desired direction while attenuating noise and interference from other directions, achieved by applying a set of weights to each sensor's signal. These weights are optimized based on the Signal-to-Noise Ratio (SNR) or desired radiation pattern.
We appreciate your insightful advice and thorough review. We hope such an explanation and revision could address the reviewer’s concern.
Comments 3: Coherency between sections should be considered.
Response 3: Thank you for pointing this out. We agree with this comment. I'm glad to hear your feedback. We have incorporated transitional sentences as suggested, enriched several content, and made corresponding modifications to enhance the logical coherence of the paper.
We hope such an explanation and revision could address the reviewer’s concern.
Comments 4: Section 4 (experiment setup) should be more obvious for readers.
Response 4: Thank you for pointing this out. We agree with this comment. We have carefully incorporated your advice by replacing the original hardware figure with a comprehensive scene figure that includes both the hardware setup and the environmental context. This modification aims to provide a clearer understanding of our experimental setup to the readers.
Additionally, we have made corresponding adjustments to the text, highlighting these modifications in red in Section 5.1.1. The modified content is as follows:
The hardware of our system is depicted in Fig. 9(a), consisting of a commercial-grade speaker and a circular arrangement of microphones. The acoustic signals emitted by the speaker are partially absorbed by the flames before being captured by the microphones.
We hope such an explanation and revision could address the reviewer’s concern.
Comments 5: A separate section related to results must be added to the manuscript. Please determine it separately.
Response 5: Thank you for pointing this out. We agree with this comment. We have carefully adjusted the structure of our manuscript to enhance the clarity of our experimental results. Specifically, we have reorganized the sections to make our work more understandable to the readers. For example, we have added Section 4 and Section 4.1 as marked in red. The modified content is as follows:
- Results
4.1. CIR measurements before and after using beamforming
We hope such an explanation and revision could address the reviewer’ s concern.
Comments 6: Conclusion needs to be rewritten. It should contain main achievements. The authors can use bullet points to clarify them.
Response 6: Thank you for pointing this out. We agree with this comment. In response to your comments, we have revised the conclusion section to more clearly articulate the key points of our work. The revised conclusion has been marked in red in the manuscript to facilitate easy identification of the changes. The revised content is shown in Page 14, the last Paragraph. The modified content is as follows:
- Conclusion
In this study, we developed and validated FireSonic, an innovative ultrasound-based system for accurately identifying indoor fire types. Key achievements include an extended sensing range of up to 7 meters, high classification accuracy of 95.5%, robust performance across various materials, resistance to environmental noise, real-time monitoring capabilities with a detection latency of less than 400 milliseconds, and the ability to identify fire types. By incorporating beamforming technology, FireSonic enhances signal quality and establishes a reliable correlation between heat release and sound propagation. These advancements demonstrate FireSonic's significant scientific and practical contributions to fire detection and safety.
We hope such an explanation and revision could address the reviewer’s concern.
Comments 7: Some English language errors should be addressed.
Response 7: Thank you for pointing out the language issues in our manuscript. We appreciate your attention to detail and understand the importance of clear and correct language in scientific communication. We carefully reviewed the manuscript and corrected the English language errors to ensure clarity and readability.
We hope such an explanation and revision could address the reviewer’s concern.
The content of the attachment is the same as that filled out on the webpage.
